# Constraints on Complex Faulting during the 1996 Ston–Slano (Croatia) Earthquake Inferred from the DInSAR, Seismological, and Geological Observations

**Marin Govorčin [1] , Marijan Herak [2], Bojan Matoš [3],\*, Boško Pribičević [1] and Igor Vlahović [3]**

1   Department of Geomatics, Faculty of Geodesy, University of Zagreb, Kačićeva 26, HR-10000 Zagreb, Croatia; marin.govorcin@geof.unizg.hr (M.G.); bosko.pribicevic@geof.unizg.hr (B.P.)

2   Department of Geophysics, Faculty of Science, University of Zagreb, Horvatovac 95, HR-10000 Zagreb, Croatia; mherak@gfz.pmf.unizg.hr

3   Department of Geology and Geological Engineering, Geology and Petroleum Engineering, Faculty of Mining, University of Zagreb, Pierottijeva 6, HR-10000 Zagreb, Croatia; igor.vlahovic@rgn.unizg.hr

\*   Correspondence: bojan.matos@rgn.unizg.hr; Tel.: +385-01-553-5787

**Abstract:** This study, involving remote sensing, seismology, and geology, revealed complex faulting during the mainshock of the Ston–Slano earthquake sequence (5 September, 1996, Mw = 6.0). The observed DInSAR interferogram fringe patterns could not be explained by a single fault rupture. Geological investigations assigned most of the interferogram features either to previously known faults or to those newly determined by field studies. Relocation of hypocentres and reassessment of fault mechanisms provided additional constraints on the evolution of stress release during this sequence. Available data support the scenario that the mainshock started with a reverse rupture with a left-lateral component on the Slano fault 4.5 km ESE of Slano, at the depth of about 11 km. The rupture proceeded unilaterally to the NW with the velocity of about 1.5 km/s for about 11 km, where the maximum stress release occurred. DInSAR interferograms suggest that several faults were activated in the process. The rupture terminated about 20 km away from the epicentre, close to the town of Ston, where the maximum DInSAR ground displacement reached 38 cm. Such a complicated and multiple rupture has never before been documented in the Dinarides. If this proves to be a common occurrence, it can pose problems in defining realistic hazard scenarios, especially in deterministic hazard assessment.

**Keywords:** southern Dinarides; Ston–Slano earthquake; earthquake relocation; DInSAR; complex faulting; coseismic deformation analysis

## 1. Introduction

The Ston–Slano earthquake sequence, with the mainshock of 5 September, 1996 ($M_w$ = 6.0, $I_{max}$ = VIII Medvedev–Sponheuer–Karnik scale, henceforth MSK), is the most important and the largest one in the southern Dalmatia zone, i.e., Dubrovnik epicentral area that occurred since the catastrophic Dubrovnik earthquake of 1667 (epicentral intensity $I_o$ = IX MSK). Described in detail by Markušić et al. [1] and Herak et al. [2], the mainshock was felt about 400 km away, and caused devastation at several localities in the epicentral area, where about 1400 buildings were damaged, of which 474 became uninhabitable [3]. Especially affected was the municipality of Ston. The old city centre suffered the most—wide cracks appeared in bearing walls (up to 10 cm wide) that also bulged and/or tilted, roofs and gables collapsed, etc. (Figure 1). Peak horizontal ground acceleration of 0.64 g recorded in Ston [4] is the largest ever observed in Croatia.

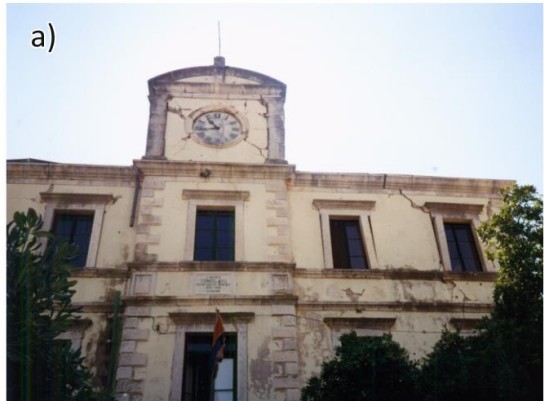
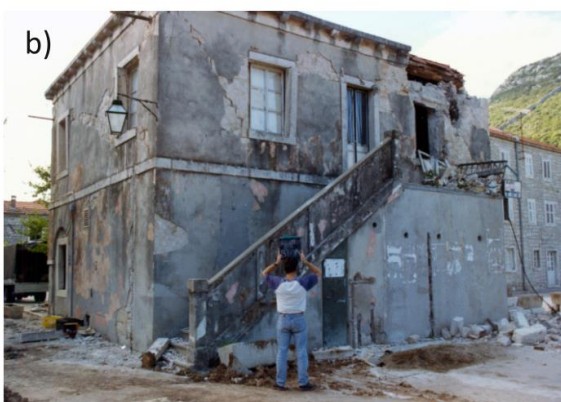
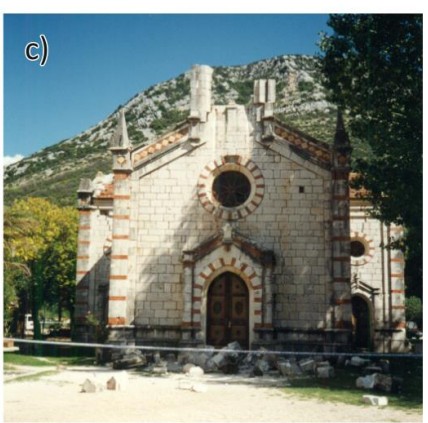
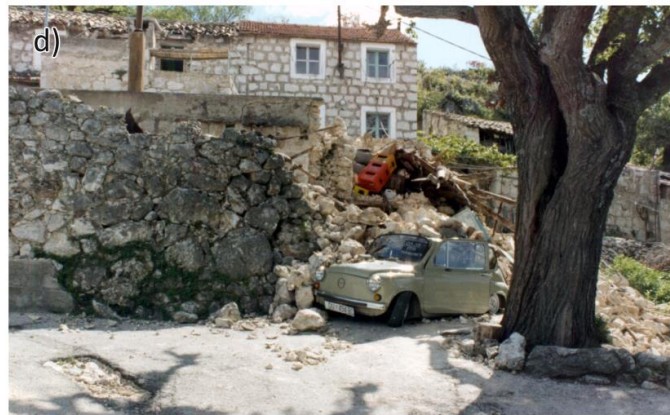

**Figure 1.** Damage in Ston ((**a**) Municipality building, (**b**) typical stone masonry building, (**c**) St. Blasius church) and in the village of Mravinca (**d**), caused by the mainshock on 5 September, 1996 (photographed by the field crew of the Department of Geophysics, Zagreb, during macroseismic survey, September 1996) [5].

The earthquake sequence lasted for about a year, with more than 1800 located aftershocks within 50 km from the mainshock's epicentre. Numerical modeling of the aftershocks' rate of occurrence was done by Herak et al. [2]. The largest aftershocks occurred on 9 September, 1996 (15:57 UTC, $M_w$ = 5.3, $I_{max}$ = VII MSK) and 17 September, 1996 (13:45 UTC, $M_w$ = 5.4).

The Ston–Slano earthquake occurred within the southern part of the Dinarides, c. 600 km long fold-and-thrust orogenic belt structurally positioned along the NE margin of the Adriatic Sea, i.e., NE part of the Adria Microplate (Figure 3). NW–SE striking Dinarides were uplifted during the Late Eocene to Oligocene due to the Adria Microplate–European Plate collision, where Adria Microplate acted as a rigid indenter ([6,7] with references therein) in respect to the European foreland (i.e., Tisza–Dacia Mega Unit; Figure 3). Southwest of the Sava Suture Zone (Figure 3), Internal Dinarides are composed of several nappe systems (Figure 3), i.e., Neotethyan ophiolitic units (Figure 3), Flysch units, and thrust sheets composed of oceanic deposits ([6,7] with references therein). The proximal part of the Adria margin is represented by External Dinaridic units (Dalmatian Zone and High Karst Unit; Figure 3), i.e., units mostly deposited on the Adriatic Carbonate Platform (see [8,9] and references therein).

The epicentral area of the Ston–Slano sequence (Figure 3) occurred within the southern Dalmatian Zone of the External Dinarides (Figures 2 and 3), very close to the tectonic contact with the High Karst Nappe towards NE (Figure 2). Dalmatian zone is composed of folded and faulted Upper Jurassic–Cretaceous carbonates deposited on the Adriatic Carbonate Platform ([8] and references therein). These carbonates are covered by Palaeogene limestones and flysch deposits (approximately 3800 m thick succession [10]). The High Karst Nappe Unit in the hanging wall of the NE inclined reverse fault is composed of c. 5000 m thick Mesozoic to Palaeogene succession (Figure 2; [10,11]). External

Dinarides are intensely folded and faulted as a result of intensive convergence of the Adria Microplate and the European Plate. Still ongoing convergence, with rates of up to 4.17 mm/yr (see [12,13] for details), is evidenced by recent seismicity in the area that accommodate stress within the collisional zone of the undeformed part of the Adriatic Microplate and the External–Internal Dinarides transition zone [14,15].

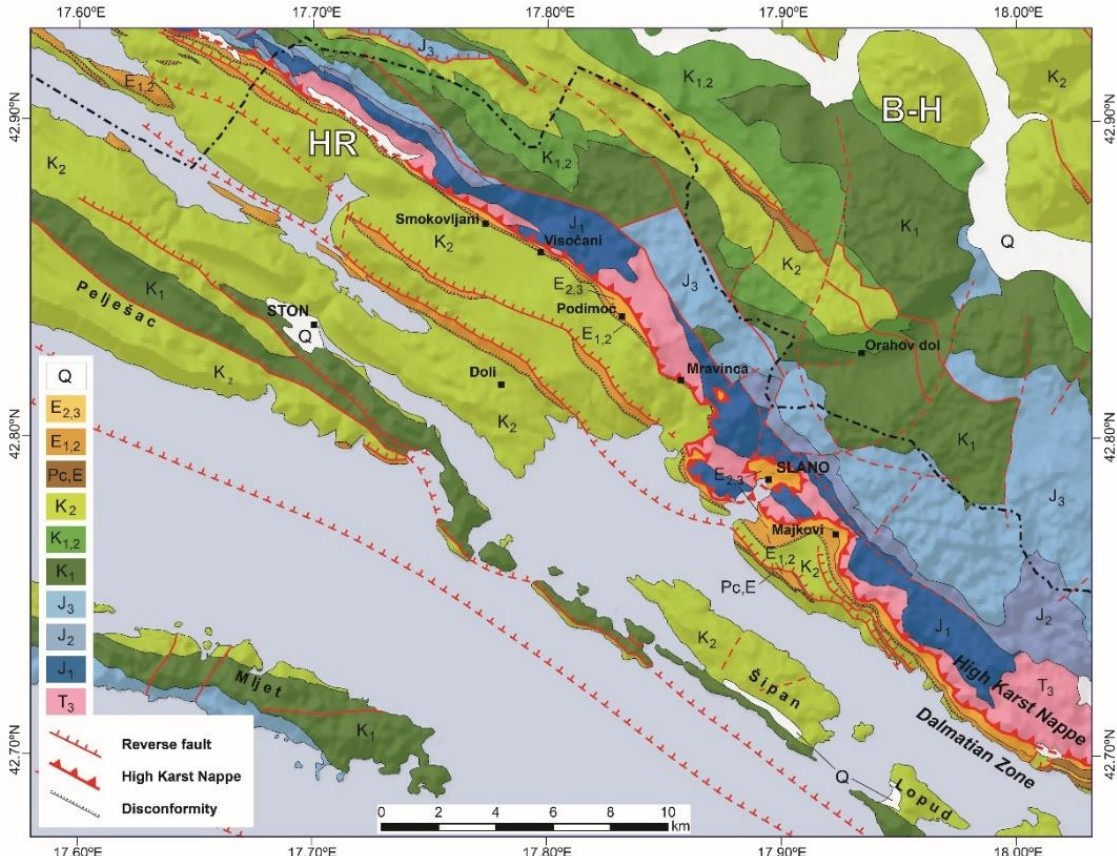

**Figure 2.** Geological map of the Ston–Slano area (simplified after [10,11]) at the NE margin of the Dalmatian Zone close to the contact with the High Karst Nappe of the External Dinarides. Legend: HR: Croatia; B-H: Bosnia and Herzegovina; $T_3$: Upper Triassic (predominantly dolomites); $J_1$, $J_2$, $J_3$: Lower, Middle, and Upper Jurassic (limestones with dolomites in the uppermost part); $K_1$, $K_{1,2}$, $K_2$: Lower, Lower/Upper transition, and Upper Cretaceous (predominantly limestones, some dolomites in the middle part); Pc, E, $E_{1,2}$: Palaeocene and Lower–Middle Eocene (foraminifera limestones); $E_{2,3}$: Middle–Upper Eocene (marls and flysch, i.e., alternation of marls and carbonate sandstones); Q: Quaternary (silts, sands, and sandstones along the coast, alluvial sandstones and gravel in the hinterland).

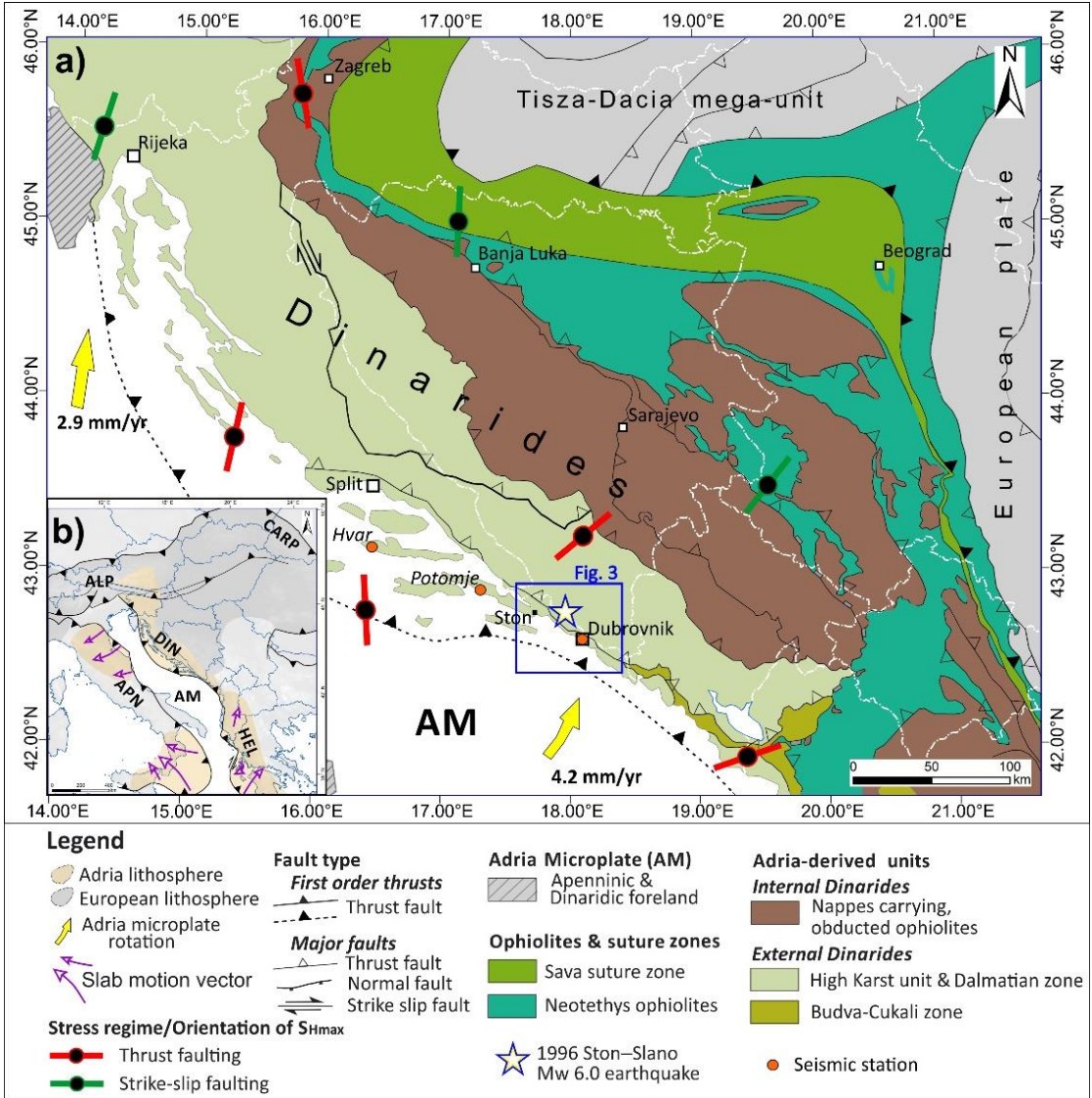

**Figure 3.** (**a**) Tectonic map of the Dinarides and surrounding areas with major tectonostratigraphic units and fault systems (after [6,9] with references). The maximum horizontal compressional stress ($S_{Hmax}$) orientations and stress regimes are based on the World Stress Map data (see [16] for details). Yellow arrows represent calculated GPS-derived velocities of the Adria Microplate in mm/yr relative to Eurasia (after [12] with references). Tectonic map indicates locations of seismic stations, location of 1996 Ston–Slano $M_w$ 6.0 earthquake, and spatial extent of Figure 2 (blue rectangle). (**b**) Tectonic sketch of the Alpine–Carpathian and circum-Adriatic orogenic system with present-day deformation fronts (e.g., major thrust and strike-slip fault systems). Adria lithosphere (beige) is subducting beneath European slab (darker grey), whereas slab motion vectors (purple arrows) indicate mantle flow (simplified after [17] with references). Abbreviations: AM: Adria Microplate; ALP: Alps; APN: Apennines; DIN: Dinarides; HEL: Hellenides.

Seismogenic sources in the southern Dalmatia zone, i.e., Dubrovnik epicentral area are predominantly NW-striking thrust faults (maximum horizontal compressional stress—$S_{Hmax}$ is NE–SW trending compression, see Figures 2 and 3) with earthquakes confined to shallow crustal levels (≤20 km in depth [18,19]). In addition, the historical seismicity (the Croatian Earthquake Catalogue [20], updated in 2019, lists seven events exceeding estimated magnitude 6.0 in the circle of 50 km radius around Dubrovnik since the 17th century), thousands of instrumentally recorded earthquakes indicate

still ongoing intense tectonic activity along the southern Dalmatian coastline within domains of the mapped faults.

Considering the fact that the study area of the southernmost Dalmatia is characterised by the highest seismic hazard in Croatia (area with several UNESCO Heritage Sites; e.g., Old City of Dubrovnik), and that the complexity and properties of seismogenic faults in the area are still not known well enough, the research objectives were to identify, and if possible, characterise source(s) of coseismic deformation related to the 1996 Ston–Slano $M_w = 6.0$ earthquake. This task is directly linked to the compilation process of the new seismic hazard map of Croatia, as fault source model is to be considered as a branch in the hazard assessment logic tree structure. In order to provide insight and effectively reinterpret local seismogenic architecture, we applied multidisciplinary approach including DInSAR to analyse the coseismic deformation in correlation with additional seismological data analysis and field structural-geological observations. Based on new findings which included relocation of macroseismic and microseismic earthquake epicentres and (re)evaluation of the Fault Mechanism Solutions (FMS), the observed DInSAR coseismic deformation, and defined structural framework of the fault system in the area, we propose principal seismogenic sources and activation scenario during the 1996 Ston–Slano earthquake sequence.

## 2. Seismological Observations

The seismological part of the study comprised the relocation of earthquakes in the greater Ston–Slano area, (re)evaluation of Fault Mechanism Solutions (FMS) for the mainshock and the strongest aftershocks, and inversion of the parameters of the macroseismic field for the mainshock of 5 September, 1996. The dataset used consisted of:

- 138,520 observed arrival times of seismic phases for 10,897 earthquakes that occurred between 1 January, 1995 and 5 September, 2018 in the area within the radius of 90 km from the mainshock of the Ston–Slano sequence, based on the seismic data of the Department of Geophysics, Faculty of Science, University of Zagreb (DGFSUZ), and supplemented by arrival times published in the ISC bulletin [21] or in available seismic bulletins from the neighbouring countries. In addition, we have also repicked some onset times from the available seismograms from the analogue and digital seismogram archive of the DGFSUZ, as well as from the international digital seismogram archives [22,23].
- A database of FMS and the corresponding first-motion polarity readings for earthquakes in Croatia and the neighbouring regions since 1910 maintained by the DGFSUZ. Those were either manually read (for all digital seismograms, and for the analogue records obtained at stations of the Croatian Seismograph Network) or adopted from various bulletins and databases. All FMS related to the studied region were rechecked and recomputed using recent crustal models.
- Intensities observed due to the Ston–Slano mainshock in Croatia and Bosnia and Herzegovina from the macroseismic archive of the DGFSUZ. The data on earthquake effects were collected by fieldwork, interviews, questionnaires sent into the greater epicentral area, and from other available sources (e.g., newspaper articles, official damage reports, etc.).

### 2.1. Earthquake Relocation

At the time of the earthquake, the seismic network of Croatia was rather sparse and equipped with analogue instruments. The two closest stations were in Dubrovnik and Hvar, about 20 and 130 km away, respectively (Figure 3). In order to better record the aftershock activity, a seismic station was deployed in Potomje on the Pelješac peninsula two days after the mainshock (Figure 3). The instrumental locations of foci for this earthquake series as found in the Croatian Earthquake Catalogue ([20], last updated in 2019; see also [1]) were determined by assuming a simple three-layered model of the crust and the uppermost mantle with horizontal interfaces. Clearly, this average model is inadequate to describe the travel times of seismic waves in a region as complex as the studied one (see

Sections 1 and 4), where Moho topography, as well as the lithology vary rapidly with the propagation azimuth and distance (e.g., [24,25]). We have, therefore, relocated all events in the 1995–2018 period using the *Hyposearch* program [26] that was recently upgraded with the option of implementing the Source-Specific Station Corrections (SSSC) [27]. The method consists of iterative locations using arrival time data corrected for the average observed residual for each specific station–phase–source triplet in the previous iteration. Such path-dependent station corrections are often used in regions of complex tectonics and geology (as is the case here) where 1D models are inappropriate to compute realistic arrival times in forward modeling, and 3D structural models do not exist (e.g., [28,29]). Each onset time used in the location was weighted depending on (i) the residual with respect to the theoretical travel-time, (ii) the wave type and the phase, (iii) the distance of the corresponding station, and (iv) the density of azimuth coverage around the station in question.

The final locations of earthquake foci were defined as the median of 20 sets of locations obtained using different models of the crust and uppermost mantle, and different configuration parameters defining computation of SSSC and weighting. The final catalogue holds basic information for 10,897 events. Figure 4 presents a subset of the most reliably located earthquakes. As can be seen from the figure, the epicentral area that was activated in the 1996–1997 sequence (Figure 4a), is virtually aseismic ever since (Figure 4b).

### 2.2. Fault Mechanism Solutions

Fault Mechanism Solutions (FMS) were determined by a grid search of fault parameters (strike, dip, rake) that best fit the observed pattern of the first P-wave motion polarity and its amplitude. In addition, we have also consulted available Centroid Moment Tensor (CMT) solutions from Istituto Nazionale di Geofisica e Volcanologia (INGV, Italy), the National Earthquake Information Centre (NEIC, USA), the International Seismological Centre (ISC, UK), and the Global CMT Catalogue [30,31]. Table 1 lists occurrence times, locations, magnitudes, and faulting parameters for the six largest events for which either a focal mechanism had been published before and/or the fault plane solution was computed in the present study. As can also be seen in Figure 5, the epicentres reported by different agencies, as well as the Best Double-Couple (BDC) of the CMT solutions or the FMS differ considerably. However, most of the solutions indicate predominance of dip-slip faulting, with some of them preferring a transpressive mechanism. This agrees well with the tectonic stress regime in this part of the External Dinarides, dominated by NE–SW oriented stress axis (Figures 3 and 5b).

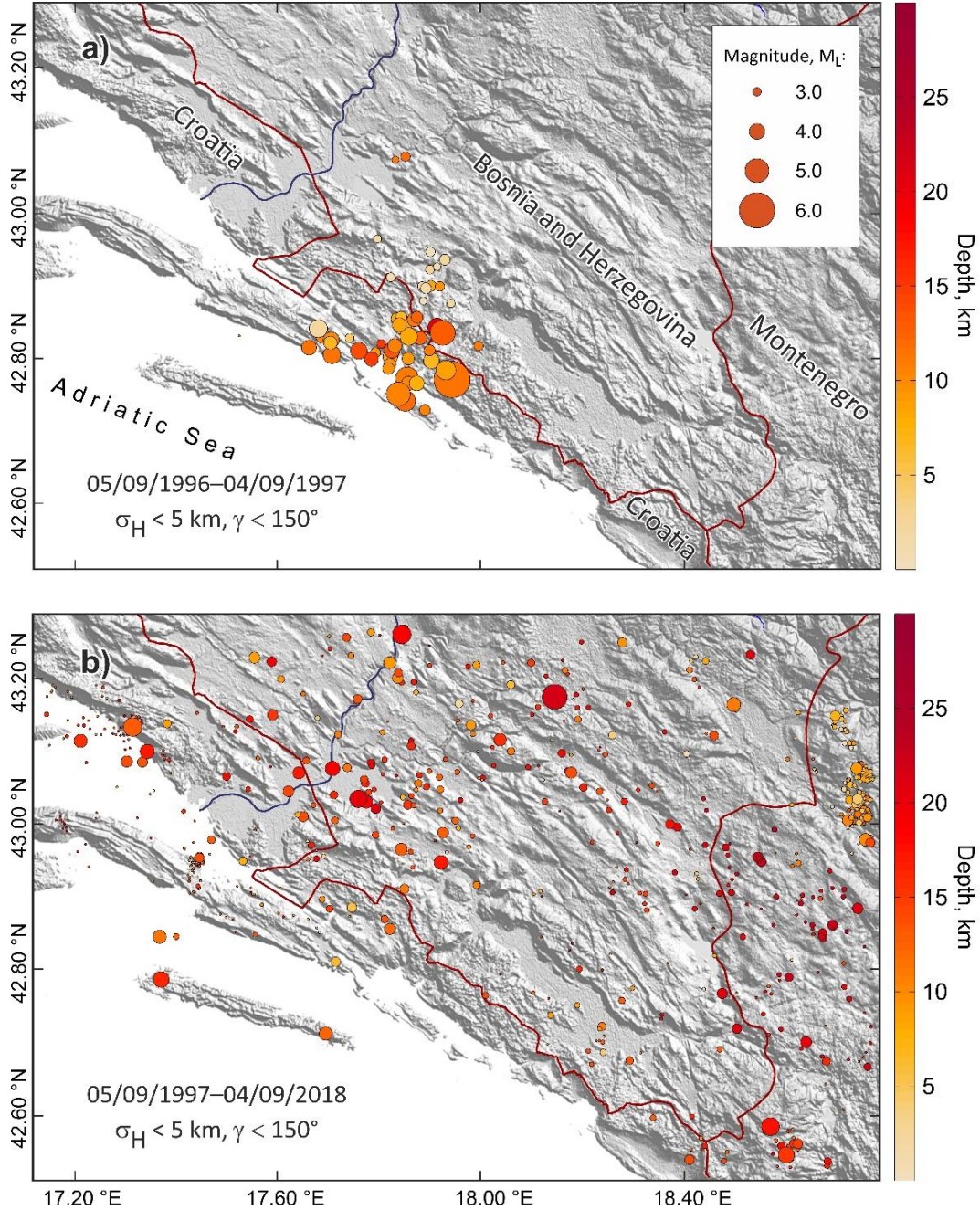

**Figure 4.** (**a**) The most reliably located epicentres in a year following the Ston–Slano mainshock. Only locations characterised by a standard error of the epicentre $\sigma_H \leq 5$ km and with azimuthal coverage characterised by a gap $\gamma \leq 150°$ are shown. Symbols are sized by the earthquake magnitude, and focal depths are indicated by the symbol colour according to the colour bar on the right. (**b**) The same as (**a**) but for the time period of 21 years, starting a year after the mainshock of 5 September, 1996.

**Table 1.** Time of occurrence, hypocentral coordinates in decimal degrees (± st. errors in km), magnitudes, and the best double-couple solutions for the mainshock and five aftershocks, as obtained by various agencies and in this study (bold). ϕ, δ, λ are the strike, dip, and rake of the best double-couple (BDC) of the CMT solution or of the FMS. [1] DT: Centroid time minus hypocentre time; C: Centroid location; E: Microseismic epicentre; BDC: Best Double-Couple; FMS: The first P-wave polarity Focal Mechanism Solution. [2] USGS [32], GCMT [33], ISC [34], ISC-GEM [35], RCMT [36].

| No. | Date | Time, DT (s)[1] | Lat., °N | Lon., °E | C, E[1] | Depth, km | Mag. | $\phi_1°, \delta_1°, \lambda_1°$ $\phi_2°, \delta_2°, \lambda_2°$ (BDC or FMS)[1] | Source[2] |
|---|---|---|---|---|---|---|---|---|---|
| 1 | Mainshock 5 September, 1996 | 20:44:17.3 DT = 8.1 s | 42.78 | 17.77 | C | 15.0 | $M_w$ 6.0, mb 5.6, $M_S$ 6.0 | 328, 32, 92 146, 58, 89 (BDC) | GCMT |
| 2 | | 20:44:09 | 42.80 | 17.94 | E | 10.0 | $M_{wc}$ 6.0 | 285, 55, 30 177, 66, 141 (FMS) | USGS |
| 3 | | 20:44:11.3 | 42.78 | 17.92 | E | 13.3 | $M_w$ 5.9 | 345, 40, 102 150, 52, 80 (FMS) | ISC |
| 4 | | 20:44:11.3 | 42.78 | 17.92 | E | 13.3 | $M_w$ 6.0, mb 5.6, $M_S$ 6.0 | 312, 69, 82 153, 23, 110 (FMS) | ISC |
| 5 | | 20:44:11 | 42.75 | 17.96 | E | 12.5 | $M_w$ 6.0 | – | ISCGEM |
| 6 | | **20:44:07.9** | **42.77 ± 5 km** | **17.94 ± 5 km** | **E** | **11.4 ± 5** | **$M_L$ 6.0** | **293, 53, 47 170, 54, 132 (FMS)** | **This study** |
| 7 | Aftershock 5 September, 1996 | 21:43:31.1 | 42.83 | 17.84 | E | 10.0 | $m_b$ 4.9 | – | RCMT (from PDE) |
| 8 | | 21:43:34.6 DT = 3.5 s | 42.74 | 17.89 | C | 15.0 | $M_w$ 4.6 | 321, 66 100 119, 26, 70 (BDC) | RCMT (from PDE) |
| 9 | | **21:43:30.3** | **42.77 ± 4 km** | **17.85 ± 4 km** | **E** | **10.8 ± 2** | **$M_L$ 4.9** | **–** | **This study** |
| 10 | Aftershock 7 September, 1996 | **05:45:32.4** | **42.85** | **17.84** | **E** | **8.1** | **$M_L$ 4.5** | **301, 59, 77 145,33, 111 (FMS)** | **This study** |
| 11 | Aftershock 9 September, 1996 | 15:57:08.7 DT = 3.6 s | 43.03 | 17.55 | C | 15.0 | $M_w$ 5.3, mb 4.8, $M_S$ 5.0 | 301, 13, 58 154, 79, 97 (BDC) | GCMT |
| 12 | | 15:57:05 | 42.77 | 17.873 | E | 10.0 | $M_{wc}$ 5.3 | – | USGS |
| 13 | | **15:57:04.6** | **42.75± 2 km** | **17.83± 2 km** | **E** | **10.5± 2** | **$M_L$ 5.0** | **223, 69, 1 133, 89, 159 (FMS)** | **This study** |
| 14 | Aftershock 17 September, 1996 | 13:45:27.9 DT = 5.1 s | 42.59 | 17.53 | C | 15.0 | $M_w$ 5.4, mb 5.4, $M_S$ 5.1 | 295, 35, 76 132, 56, 99 (BDC) | GCMT |
| 15 | | 13:45:22 | 42.87 | 17.82 | E | 10.0 | $M_{wc}$ 5.4 | – | USGS |
| 16 | | **13:45:22.1** | **42.83 ± 2 km** | **17.92 ± 2 km** | **E** | **12.8 ± 1** | **$M_L$ 5.1** | **299, 33, 77 134, 58, 98 (FMS)** | **This study** |
| 17 | Aftershock 20 October, 1996 | **15:00:01.7** | **42.78 ± 1 km** | **17.93 ± 1 km** | **E** | **8.7 ± 2** | **$M_L$ 4.6** | **319, 47, 83 149, 43, 97 (FMS)** | **This study** |

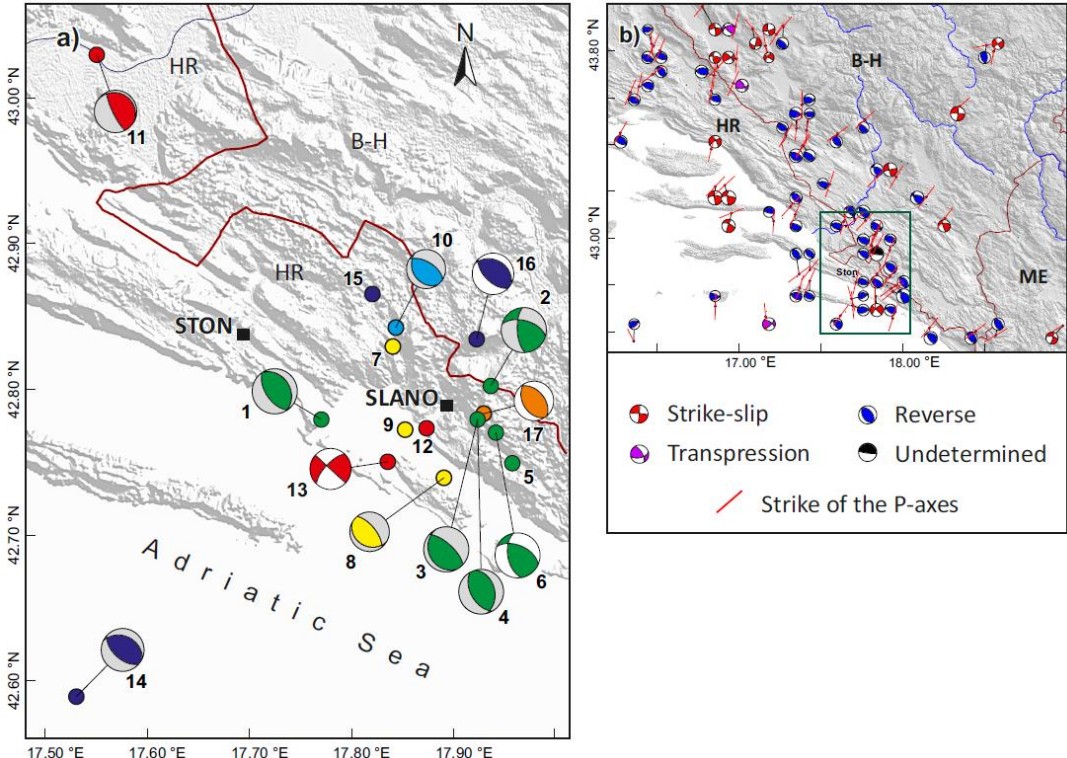

**Figure 5.** (**a**) Epicentres and fault-plane solutions (lower hemisphere projections, coloured compressive quadrants) reported in Table 1. The same colour refers to the same event. The numbers refer to entries in Table 1. White and grey dilatational quadrants denote focal mechanism solutions (FMS) and the best double-couple (BDC) of the centroid moment tensor (CMT) solutions, respectively (see Table 1). HR: Croatia; B-H: Bosnia and Herzegovina. (**b**) All FMS from the database of Department of Geophysics, Faculty of Science, University of Zagreb (DGFSUZ). Green rectangle shows the area shown in part (**a**). HR: Croatia; B-H: Bosnia and Herzegovina; ME: Montenegro.

### 2.3. Inversion of the Macroseismic Field

Macroseismic investigations started immediately after the mainshock occurred. The collected data (see above) permitted assigning intensity to 145 localities. Macroseismic field of the mainshock was modeled using the *MEEP* v.2.0 algorithm [37] modified as described in detail by [38,39]. Figure 6 presents Intensity Data Points (IDPs) in the epicentral area. Maximum intensity of VIII MSK was observed in Ston and in the villages of Podimoć and Mravinca (Figure 1). We have inverted all IDPs with intensities $I_{obs} \geq$ VI MSK for the macroseismic earthquake parameters: Coordinates of the macroseismic epicentre ($\phi_m$, $\lambda_m$), macroseismic focal depth ($h_m$), and epicentral intensity ($I_0$). The obtained parameters are:

$$\phi_m = 42.825\,^\circ\text{N} \pm 3\,\text{km}, \lambda_m = 17.835\,^\circ\text{E} \pm 4\,\text{km}, h_m = 6 \pm 2\,\text{km}, I_0 = 8.1\,\text{MSK}$$

In order to homogenise the dataset and reduce observations to the average soil, in the inversion we have reduced the observed intensity in Ston by half a degree of MSK (from VIII to VII–VIII MSK), since Herak et al. [4] have shown that the damage to the building stock in the old town centre was closely related to the estimated soil amplification determined by ambient noise measurements. The macroseismic epicentre is located close to the Podimoć village, about 10.5 km to the NW from the microseismic epicentre where the fault rupture started (Figure 6).

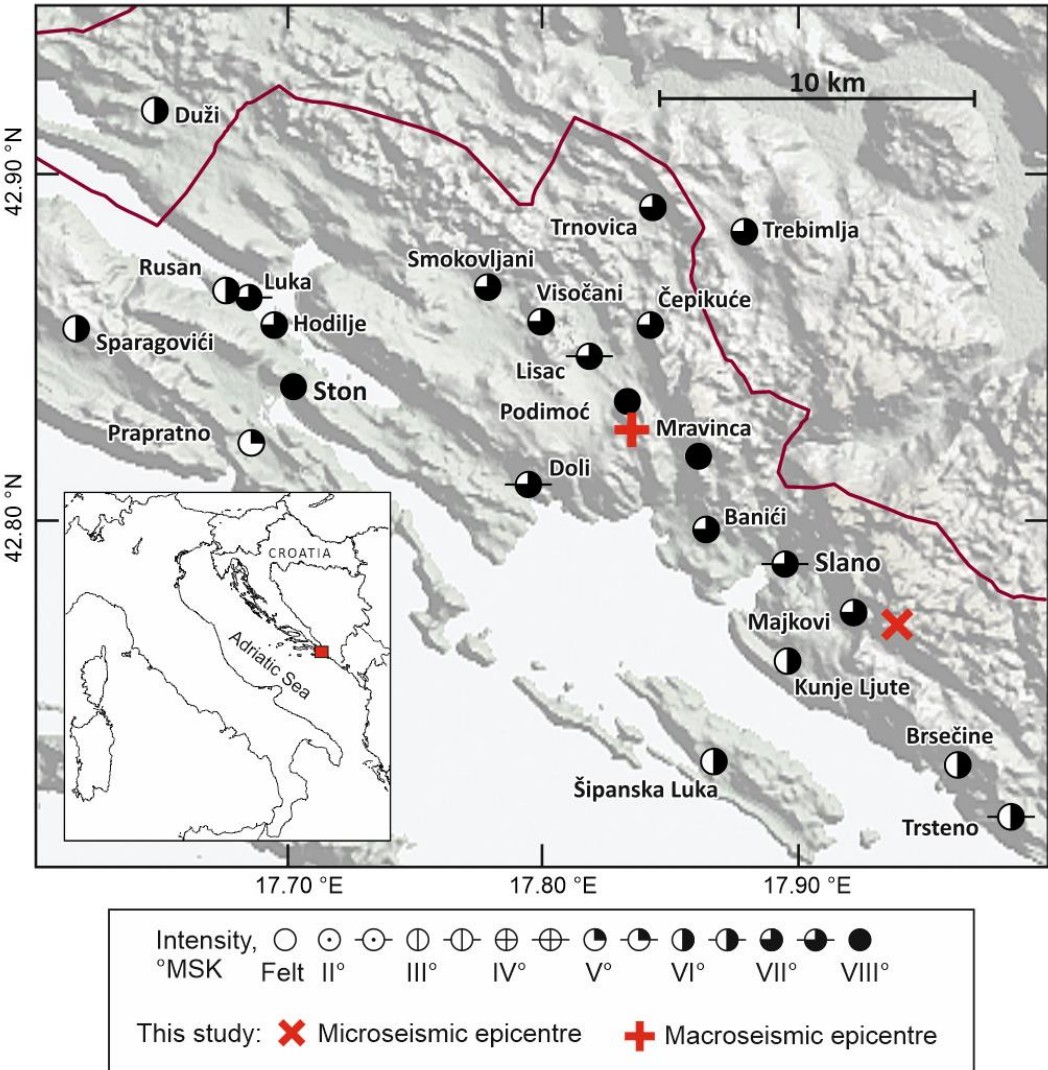

**Figure 6.** Macroseismic map of the epicentral area of the Ston–Slano mainshock, 5 September, 1996. Intensities according to the Medvedev–Sponheuer–Karnik (MSK) scale. The inset shows the geographical position of the area shown.

## 3. DInSAR Observations

The Differential Interferometric Synthetic Aperture Radar (DInSAR) technique has been widely used for investigation and characterisation of coseismic surface displacements [40]. The technique is based on the measurement of the differential phase angle between two coherent Synthetic Aperture Radar (SAR) acquisitions over the same area. After subtraction of the topographic phase component from the obtained phase difference, the technique can determine the ground surface motion caused by the earthquake in the radar Line-Of-Sight (LOS) direction.

In this study, we used the SAR images acquired by the C-band (5.66 cm wavelength) European Remote Sensing satellite (ERS-2) before and after the earthquake from both ascending and descending orbits. The coseismic interferograms were formed from the image pairs described in Table 2 by using InSAR Scientific Computing Environment (ISCE) software developed by the Jet Propulsion Laboratory (JPL; [41]). Coseismic interferograms were generated from the ERS-2 Single-Look Complex (SLC) products obtained from the European Space Agency (ESA). We selected the interferometric pairs with a favourable perpendicular baseline ($B_{perp}$) smaller than 60 m to suppress the topographic phase contribution in the interferometric phase. The height of ambiguity of the ascending and descending

ERS-2 interferometric pairs gives a $1.8 \times 10^{-2}$ cm and $5.1 \times 10^{-3}$ cm error in ground displacement determination with a 1 m change in topography, respectively. This implies a low sensitivity of the interferometric phase to topographic errors, typically on the order of ~10 m for the Shuttle Radar Topography Mission (SRTM) Digital Elevation Model (DEM) [42]. The Ston–Slano area represents a perfect location for DInSAR application due to infrequent rainfall, as well as sparse vegetation and human cultivation of land, meaning that the long-wavelength tectonic signal can be observed even after a long time. However, the significant atmospheric signal contamination of the interferometric phase (erroneous phase change due to perturbations in pressure, temperature, and relative humidity in the lower part of the troposphere, < 5 km) can be still found especially near the coast. In the selection process of the best interferometric pairs, we paid special attention to the potential atmospheric phase contamination by using the pairwise logic [43]. For example, we discarded the interferometric pair between 9 August, 1996 and 27 July, 1997 from descending orbit with a $B_{perp}$ of –41 m from this study, as we found that the phase of SAR image acquired on the 27 July, 1997 was strongly affected by atmospheric noise.

**Table 2.** Interferometric pairs used in differential interferometric synthetic aperture radar (DInSAR) processing (mainshock occurred on 5 September, 1996). $B_{perp}$ and $B_{temp}$ represent perpendicular and temporal baseline of DInSAR pairs respectively, whereas $H_{amb}$ is height of ambiguity.

| Sat. | Orbit Direction | Track | Master Image (yr/mm/dd) | Slave Image (yr/mm/dd) | $B_{perp}$ (m) | $B_{temp}$ (days) | $H_{amb}$ (m) |
|---|---|---|---|---|---|---|---|
| ERS-2 | Descending | 451 | 09/08/1996 | 16/051997 | –59 | 280 | 158 |
| ERS-2 | Ascending | 501 | 06/11/1995 | 01/091997 | –17 | 665 | 547 |

The DInSAR processing can be divided into several steps: coregistration, interferogram generation, flat-phase, and topographic phase contribution corrections, adaptive filtering, phase unwrapping, and geocoding. More details about DInSAR processing steps can be found in Hanssen [44]. We applied the precise orbital information provided by the Delft Institute for Earth-Oriented Space Research in the Netherlands [45] for the coregistration of the SAR SLC images and flat-phase contribution removal from the interferometric phase. Tropospheric phase correction was applied by removing the tropospheric phase contribution simulated on the 1 arc s (30 m) SRTM DEM. After tropospheric phase correction, an adaptive Goldstein filter [46] with alpha value of 0.7 was used to further reduce the interferogram phase noise. The obtained filtered interferometric phase wrapped by the $2\pi$ moduli was unwrapped with a minimum cost-flow SNAPHU algorithm [47] to obtain relative LOS surface displacements and then geocoded to a WGS84 geographic coordinate system. Due to relatively high coherence (0.65 ± 0.24; Figure 8b), the obtained interferograms are considered to be reliable. We applied a multilook ratio of 1:5 for range and azimuth directions, respectively, to obtain ~20 m pixel posting of the geocoded unwrapped interferograms. The final results are the geocoded coseismic surface displacement fields in the LOS directions of ERS-2 ascending and descending orbits (see Figures 7 and 8). The displacement fields represent LOS surface movements with respect to the reference point (unwrapping seed point), located arbitrarily outside of the deformation zone.

Figure 7a,c shows a very spatially complex coseismic displacement field with several sets of apparently overlapping concentric fringe patterns, which indicates that multiple faults and/or fault segments may have ruptured during the earthquake sequence. Both interferograms include the postseismic period comprising two $M_w > 5.0$ aftershocks (Table 1 and Figure 5a) that could have interfered with the mainshock's displacement field.

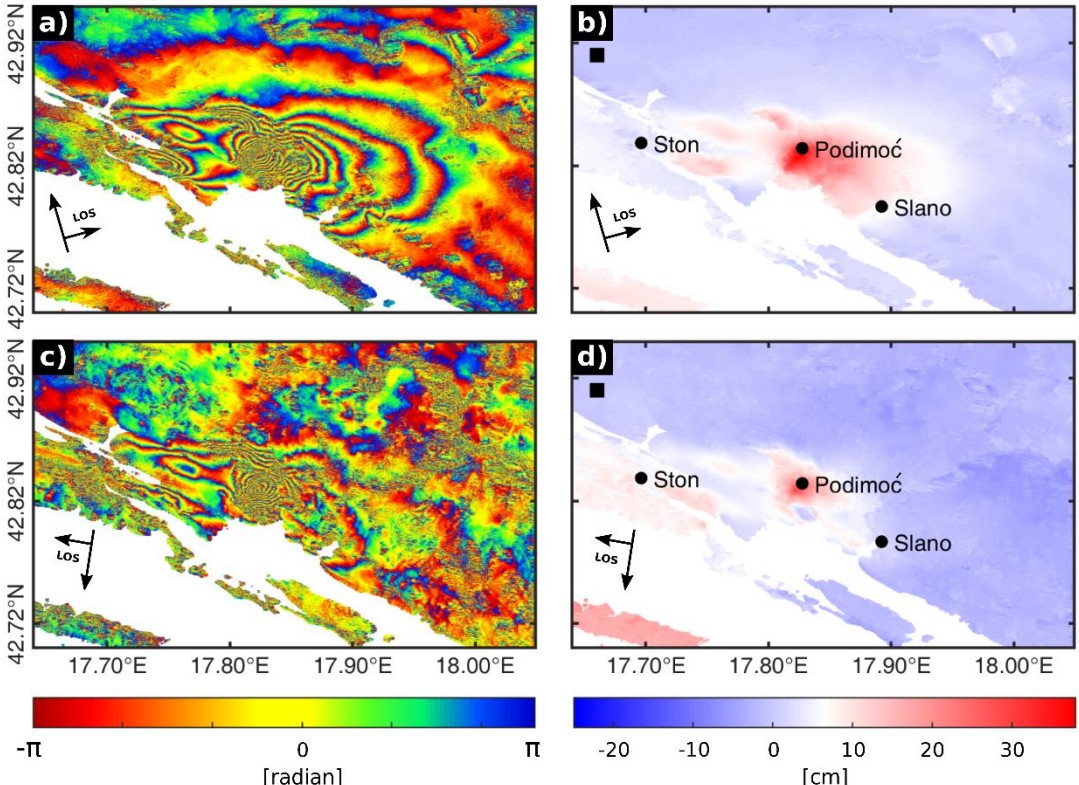

**Figure 7.** (**a**) European radar satellite (ERS-2) ascending orbit wrapped interferogram of the coseismic displacement induced by the Ston–Slano earthquake. (**b**) ERS-2 ascending orbit unwrapped interferogram of the coseismic displacement. (**c**) ERS-2 descending orbit wrapped interferogram of the coseismic displacement. (**d**) ERS-2 descending orbit unwrapped interferogram of the coseismic displacement. Line-of-sight (LOS) surface displacements in (**a**) and (**c**) are wrapped by the modulus $2\pi$, where one colour cycle represents ground motion of 2.8 cm (the satellite's beam half wavelength). Black squares in the upper left corners mark the reference point for the phase unwrapping located outside of the deformation zone.

The fringe patterns show the ground movement towards the satellite in the LOS directions that spans over the area about 19.2 km by 11.6 km extending NW from Slano. The main difference between the ascending (Figure 7a) and the descending (Figure 7c) orbit interferograms is a clear absence of the fringe (pattern P1a, Figure 8a) in the descending one. This is consistent with the suggestion that the coseismic ground displacement was generated by a transpressional left-lateral fault with a large reverse component (e.g., FMS No. 2 and 6 in Table 1 and Figure 5a) dipping to the NE, in which the hanging wall of the causative fault moves up and to the NW, towards the satellite in its ascending trajectory (Figure 7a). The maximum observed LOS displacement of ~38 cm (Figure 7b) and ~30 cm (Figure 7d) is found near the Podimoć village (pattern P5, Figure 8a) in the ascending and descending interferogram, respectively. Its location lies about 11 km to the NW of the microseismic epicentre determined here (No. 6 in Table 1 and Figure 5a), but coincides with the inverted position of the macroseismic epicentre (Figures 5 and 7), i.e., the location of maximum damage. Although half of pattern P5 is decorrelated (Figure 8b), the other half shows a high fringe gradient (Figure 8a) that suggests that most of the slip occurred on a shallow fault segment extending to the surface with a possible surface rupture associated with the observed decorrelation.

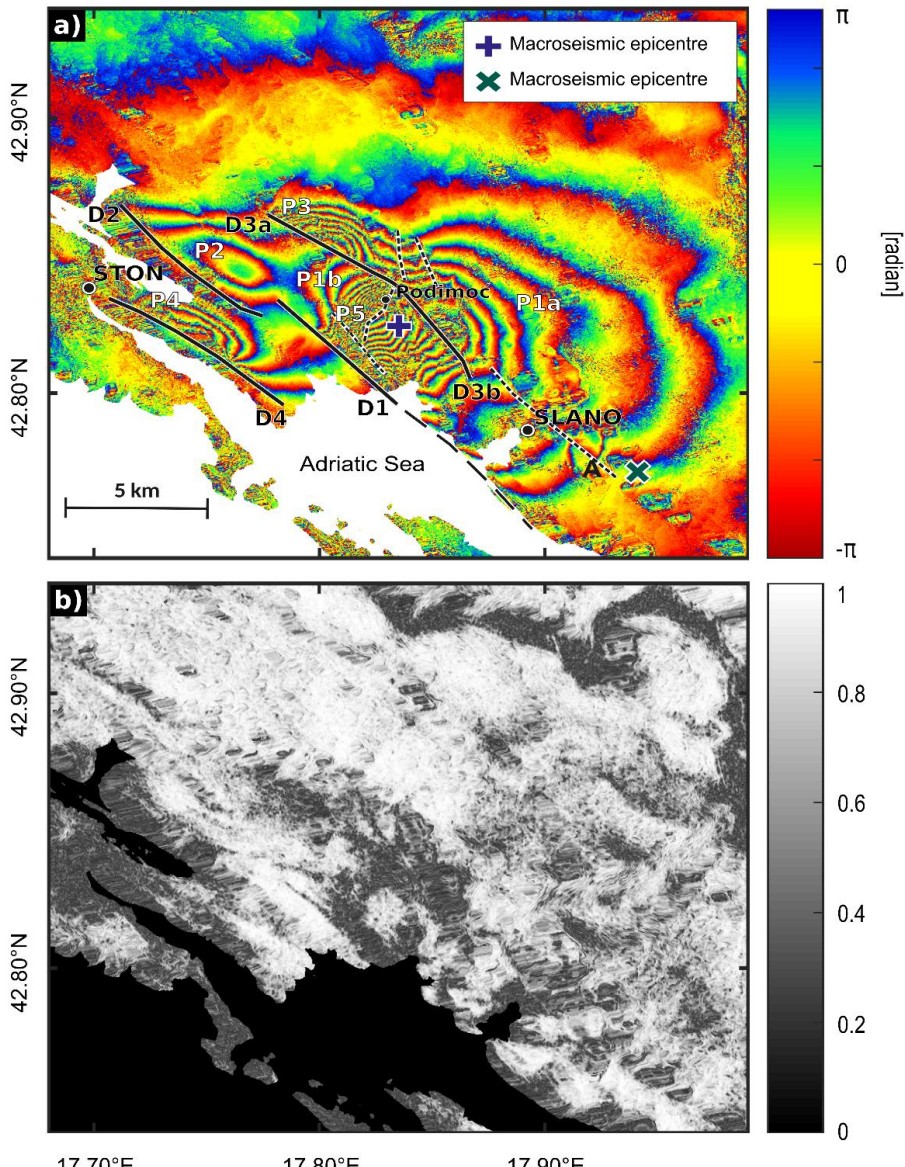

**Figure 8.** (**a**) Black lines (D1–D4) lie along the fringe discontinuities observed in both interferograms (Figure 7a,c; only the ascending interferogram is shown here). Long, dashed line indicates assumed continuation of the discontinuity offshore. Short dashes emphasise less clearly expressed, secondary discontinuities. None of the observed discontinuities correlates with topography. White tags P1–P5 specify the fringe patterns. The blue and green crosses mark the macroseismic and microseismic epicentre of the mainshock, respectively. 'A' marks the pattern deformation close to the microseismic epicentre. (**b**) Coherence of ERS-2 ascending orbit unwrapped interferograms with value range (0.65 ± 0.24).

## 4. Geological Observations

The structural-geological analysis of the Ston–Slano area in terms of fault kinematics determination, i.e., relation between observable fault structures in respect to the past and present stress fields, was mainly focused along the mapped faults at the contacts and within the Mesozoic and Eocene successions, areas of relocated microseismic and macroseismic epicentres, as well as some specific DInSAR observations (indicating previously unknown surface discontinuities, e.g., discontinuity D4; Figures 2, 8 and 9). In the studied area (about 25 km long and 10 km wide; Figure 9) structural data were collected

on outcrop-scale fault planes (e.g., dip direction/angle of fault planes and orientation of carbonate slickensides including their azimuth/plunge and sense of movement). Collected structural data for 105 fault planes on 129 locations (Figure 9) were separated into kinematically compatible datasets and processed by Tectonics FP software [48]. Using the P–T axis method [49,50] the theoretical maximum (σ1), intermediate (σ2), and minimum stress axes (σ3) were calculated, whereas palaeo-synthetic focal mechanisms for analysed fault segments were determined as representatives of the palaeostress fields with the Right Dihedra Method [51].

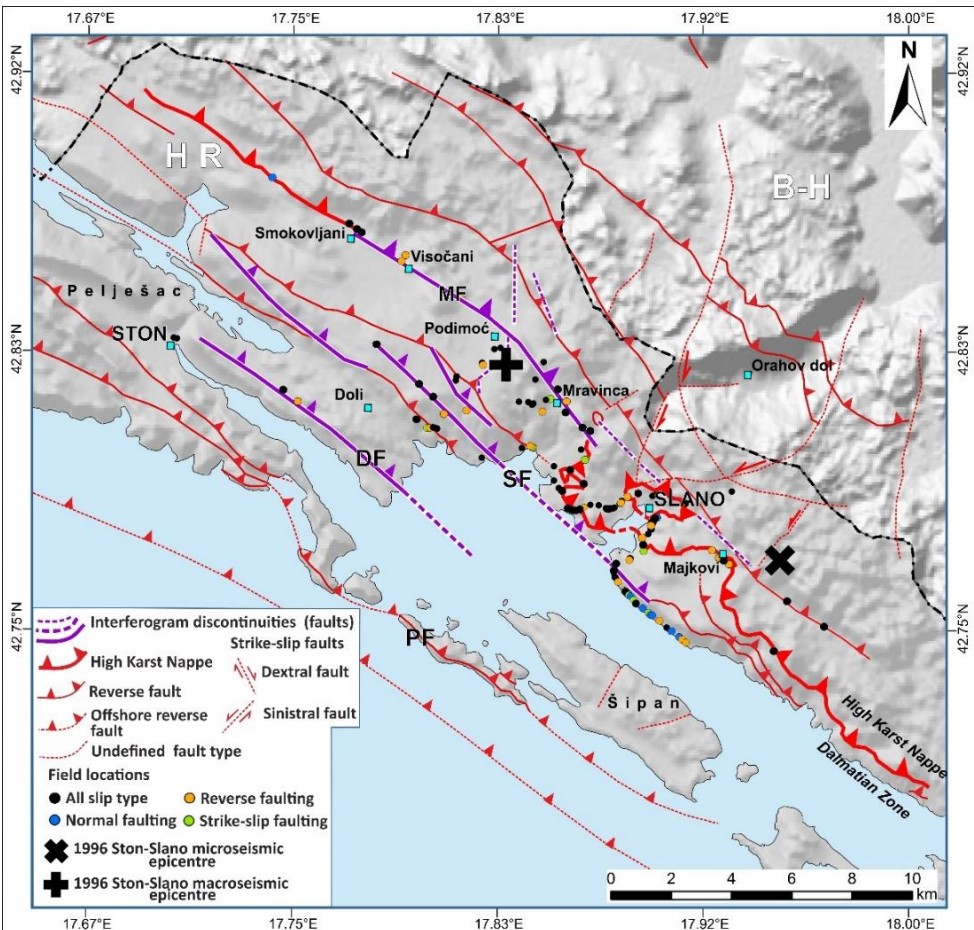

**Figure 9.** Simplified structural map of the Ston–Slano area with faults mapped onshore and offshore. Fault traces are generalised after [10,11,19]. Fault measurements conducted at the 127 locations are indicated with multicolour dots in accordance to the measured fault's kinematic properties. Locations of microseismic and macroseismic epicentres of the 1996 Ston–Slano earthquake are indicated in accordance to Figure 2, whereas purple lines are the fringe discontinuities interpreted in Figure 8. Abbreviations: PF: Pelješac fault; DF: Doli fault; SF: Slano fault; MF: Mravinca fault.

Structural measurements of the mapped fault planes within the Ston–Slano area indicated both dip-slip and strike-slip fault kinematics, whereas observed oblique-slip kinematics were characterised by a dominant reverse dip-slip component. Positioned in heavily tectonised zones, mapped fault planes were separated into four compatible structural datasets (Figure 3). Accordingly, based on the kinematic criteria, 83 measured fault planes were characterised by dip-slip kinematics, out of which 59 have reverse and 24 normal faulting kinematics. Twenty-two fault planes accommodated either oblique-slip or strike-slip motion. The aforementioned fault kinematic groups were subdivided into compatible fault groups and fault group subsets (Figure 10 and Table 3).

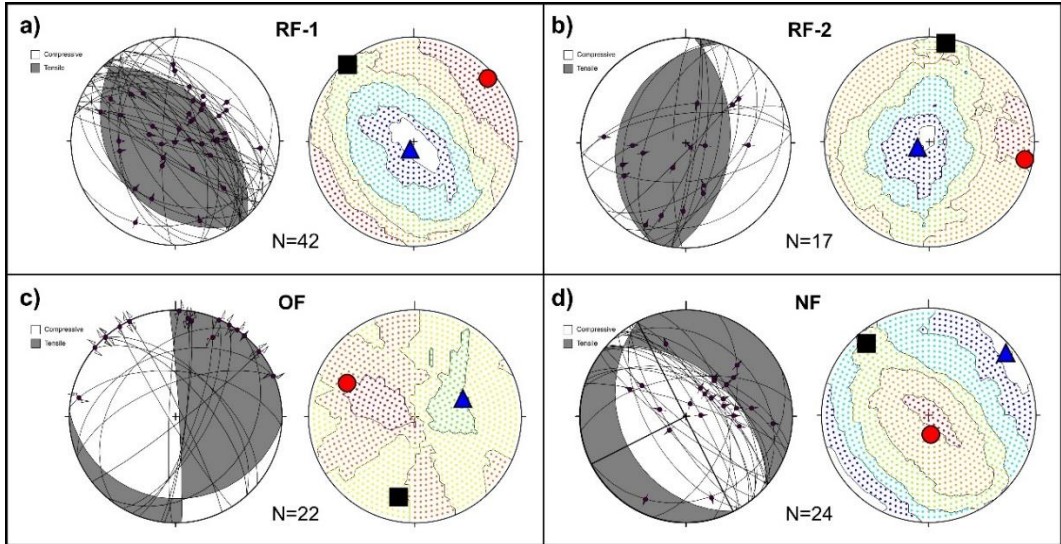

**Figure 10.** Structural diagrams for the Ston–Slano fault zone. The white quadrants on the structural beach-ball diagrams represent compression, while the shaded quadrants represent tension. (**a**) Reverse fault group 1 (RF-1); (**b**) Reverse fault group 2 (RF-2); (**c**) Oblique fault group (oblique-slip or strike-slip)(OF); (**d**) Normal fault group (NF). The red dots, black rectangles, and blue triangles indicate $\sigma_1$, $\sigma_2$, and $\sigma_3$, respectively.

**Table 3.** Mean geometric properties of the observed fault planes within the Ston–Slano fault zone with calculated kinematic indicators and parameters. Fault planes were delineated with respect to their geometrical properties and kinematic compatibility within the following groups (see Figure 10): RF-1: Reverse fault group 1; RF-2: Reverse fault group 2; OF: Oblique fault group (oblique-slip or strike-slip); and NF: Normal fault group. Fault types: R: Reverse; O-S: Oblique-slip; S-S: Strike-slip; N: Normal. Orientation of the P- and T-axes are based on constructed synthetic structural beach-ball diagrams.

| Fault group | Fault subset | No. of fault data | Dip azimuth (°) | Dip angle (°) | Pitch (°) | Fault type | Striation Trend (°) | Striation Plunge (°) | P-axis Trend (°) | P-axis Plunge (°) | T-axis Trend (°) | T-axis Plunge (°) |
|---|---|---|---|---|---|---|---|---|---|---|---|---|
| RF-1 | RF-1a | 27 | 44 | 51 | 69 | R | 113 | 49 | 50 | 8 | 205 | 82 |
| | RF-1b | 15 | 216 | 32 | 69 | | 237 | 49 | | | | |
| RF-2 | RF-2a | 11 | 95 | 52 | 57 | R | 139 | 46 | 100 | 8 | 247 | 80 |
| | RF-2b | 6 | 291 | 32 | 51 | | 218 | 46 | | | | |
| OF | OFa | 16 | 86 | 88 | 3 | O-S | 138 | 4 | 296 | 30 | 70 | 50 |
| | OFb | 6 | 184 | 20 | 0 | S-S | 100 | 1 | | | | |
| NF | NFa | 18 | 41 | 51 | 69 | N | 85 | 51 | 173 | 75 | 51 | 08 |
| | NFb | 6 | 243 | 4 | 55 | | 221 | 49 | | | | |

Results of structural analysis show that reverse fault planes can be separated into two fault groups and four subsets. The first reverse fault group (RF-1; Figure 10a) is characterised by conjugate fault pairs, i.e., fault subsets characterised by the NW-SE strike, dipping both towards the NE and SW (dip angles are 51 and 32°, respectively; Table 3), indicating NE- and SW-directed tectonic transport (Figures 10a and 11). Structural analysis of the palaeostress field using the P–T axis method [50,51] and derived synthetic structural focal mechanisms (Figure 10a), indicated that the observed palaeostress compressional field is associated with a P-axis predominantly trending NE-SW (see Table 3 for details).

The second group of the observed reverse faults (RF-2; Figure 10b) is generally characterised by conjugate fault pairs dipping towards ESE and WNW and WNW- and ESE-directed tectonic transport, respectively. Computed representative palaeostress field indicates a compressional palaeostress field associated with a P-axis dominantly trending NW-SE (Figure 10b and Table 3). In addition to typical compressional structures, i.e., reverse fault planes, 22 oblique-slip or strike-slip fault planes were measured in this study and were grouped into the oblique-slip fault group (OF; Figure 10c). With dominant N–S (locally NW–SE and NE–SW) and E–W strike (see Figure 10c and Table 3) and subvertical to subhorizontal geometry, respectively, observed subvertical and sinistral faults are kinematically linked with prevalence of transpressional tectonic phase (Figure 10c) characterised by NW–SE trending P-axis that gently dips (30°) towards NW (see Figure 10c and Table 3 for details).

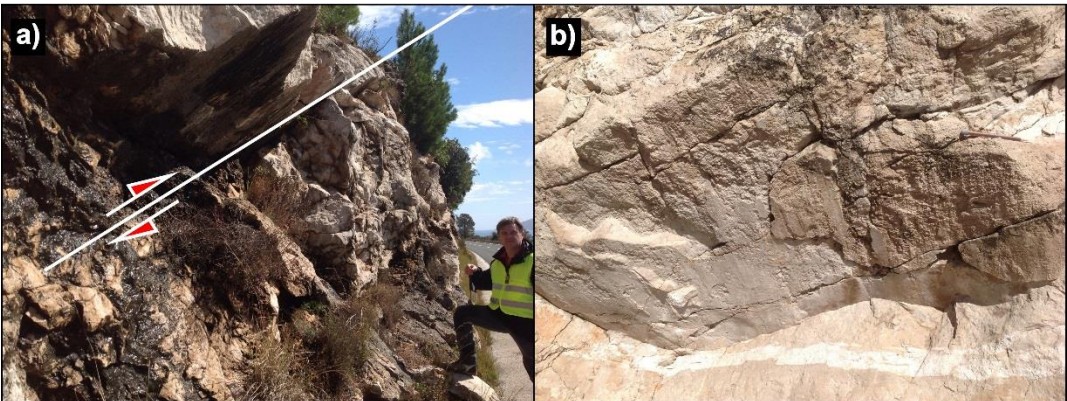

**Figure 11.** Photograph (**a**) of the reverse fault plane (Fp = 16/55; indicating SW-directed tectonic transport) with detail of visible reverse motion indicators, i.e., fault striations (**b**) within the Upper Cretaceous carbonates c. 3 km SW of Slano (Date: 7 March, 2018; Location: 42.76 °N, 17.88 °E; Photographed by: Bojan Matoš).

In the Ston–Slano area, within the mapped fault zones 24 normal fault planes were measured (NF; Figure 10d). Observed NW–SE striking fault subsets are dipping both towards NE an SW, at the dip angles of 51 and 34°, respectively (Figure 10d; Table 3). At the same time, structural analysis of the palaeostress field, i.e., P–T axis method, as well as derived synthetic structural focal mechanisms (Figure 10d and Table 3) indicated a locally expressed extensional kinematic tectonic phase in the study area. This tectonic phase was associated to subvertical P-axis (orientation of P-axis is 173/75; see Table 3) which resulted in the general E–W local extension of the observed structures.

## 5. Discussion

As presented above, the observed coseismic displacement field of the Ston–Slano earthquake sequence is surprisingly complex. It suggests that at least four, but possibly as many as nine distinct fault segments ruptured during the sequence (Figure 8a). It is unfortunate that no coherent DInSAR interferograms generated with a second SAR image acquired shortly after the mainshock could have been obtained. The only, marginally useful interferogram, is made using a SAR acquisition from the descending orbit of 13 September, 1996 (Figure 12a), i.e., eight days after the mainshock. Comparing it to Figure 12b (also an interferogram generated with SAR acquisitions from the descending orbit), it is clear that nearly all main features of the coseismic surface deformation are already present on the ERS-2 acquisition on 13 September, 1996. This implies that, for instance, the strongest aftershock (17 September, 1996, 13:45, $M_w$ 5.4, see Table 1) had practically no role in formation of the final pattern of the coseismic surface deformation obtained after the earthquake sequence ended, i.e., observed in the interferograms in Figure 7. In the period before 13 September, 1996 there was only one event that could have caused ground deformation large enough to be measured with DInSAR—the one of 9 September, 1996 ($M_w$ 5.3, see Table 1). The expected fault rupture length and rupture area for a magnitude $M_w$ 5.3

earthquake are 3–5 km and 15–20 km$^2$, respectively (according to [52]), so it is possible that at least one of the observed fringe discontinuities shown in Figure 8a is due to the aftershock of 9 September, 1996. Given the position of its epicentre (Table 1 and Figure 5) and taking into account rather large confidence intervals for its best location, the most likely candidate is pattern P4 that is related to the Doli fault (Figures 8a and 9). However, as even stronger aftershock of 17 September, 1996 apparently did not cause ground displacements observable with DInSAR, and knowing that recent events of this size in the greater area were not recognized in the DInSAR interferograms, this will have to remain an unsolved issue.

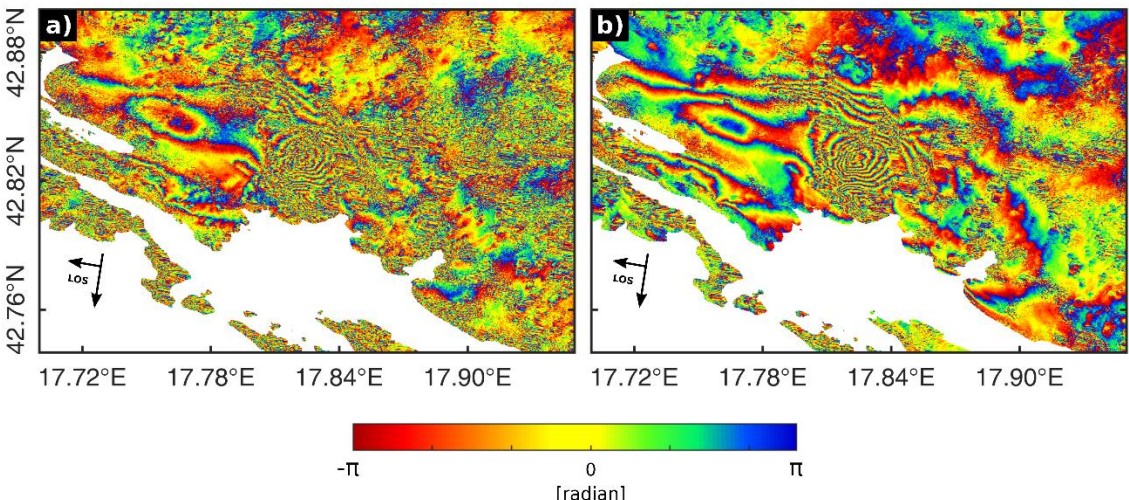

**Figure 12.** ERS-2 descending orbit wrapped interferograms of the coseismic displacement obtained from interferometric pair; (**a**) 25 August, 1995–13 September, 1996 (B$_{perp}$ 108 m); (**b**) 9 August, 1996–16 May, 1997 (B$_{perp}$ -59 m). LOS surface displacements are wrapped by the modulus $2\pi$, where one colour cycle represents ground motion in a size of the satellite's beam half wavelength (2.8 cm).

Taking all stated above into consideration, we are left with a case of apparently very complex multiple fault rupture during the main event of 5 September, 1996. It started about 4.5 km ESE of Slano near the Majkovi village (Figure 6), at the depth of about 11 km (Table 1), and proceeded NW-wards, either along the reverse Slano fault (SF) or the Pelješac fault (PF) (Figures 8a, 9 and 13) whose traces are mapped about 4 and 9 km to the SSW, respectively. Both of their strikes agree well with the strike of one of the nodal planes of the FMS ($\phi_1 = 293°$, see Table 1). Assuming the dip of the fault as $\delta_1 = 53°$ and a focal depth of 11.4 ± 5 km (Table 1), and also taking the uncertainty of the epicentral location (± 5 km) of the FMS parameters into account, neither of the faults can be given a definite preference. However, inspection of Figure 8a hints that the fringes of the P1a and P2 patterns stop at the discontinuities D1 and D2, which corresponds to the Slano fault as mapped in the field (Figures 8a and 9), so we suggest that it is probably the Slano fault where the main rupture started. The near-field coseismic ground deformation in the immediate vicinity of the epicentre is clearly seen in the ascending orbit interferogram (Figure 8a, mark 'A'). Left-lateral transpressive rupture proceeded NW-wards for about 8 km (dashed part of D1 in Figure 8a), producing the main lobes P1a (Figure 8a). Their wide spatial extent in the hanging wall suggests that the rupture was confined to the deep parts of the fault.

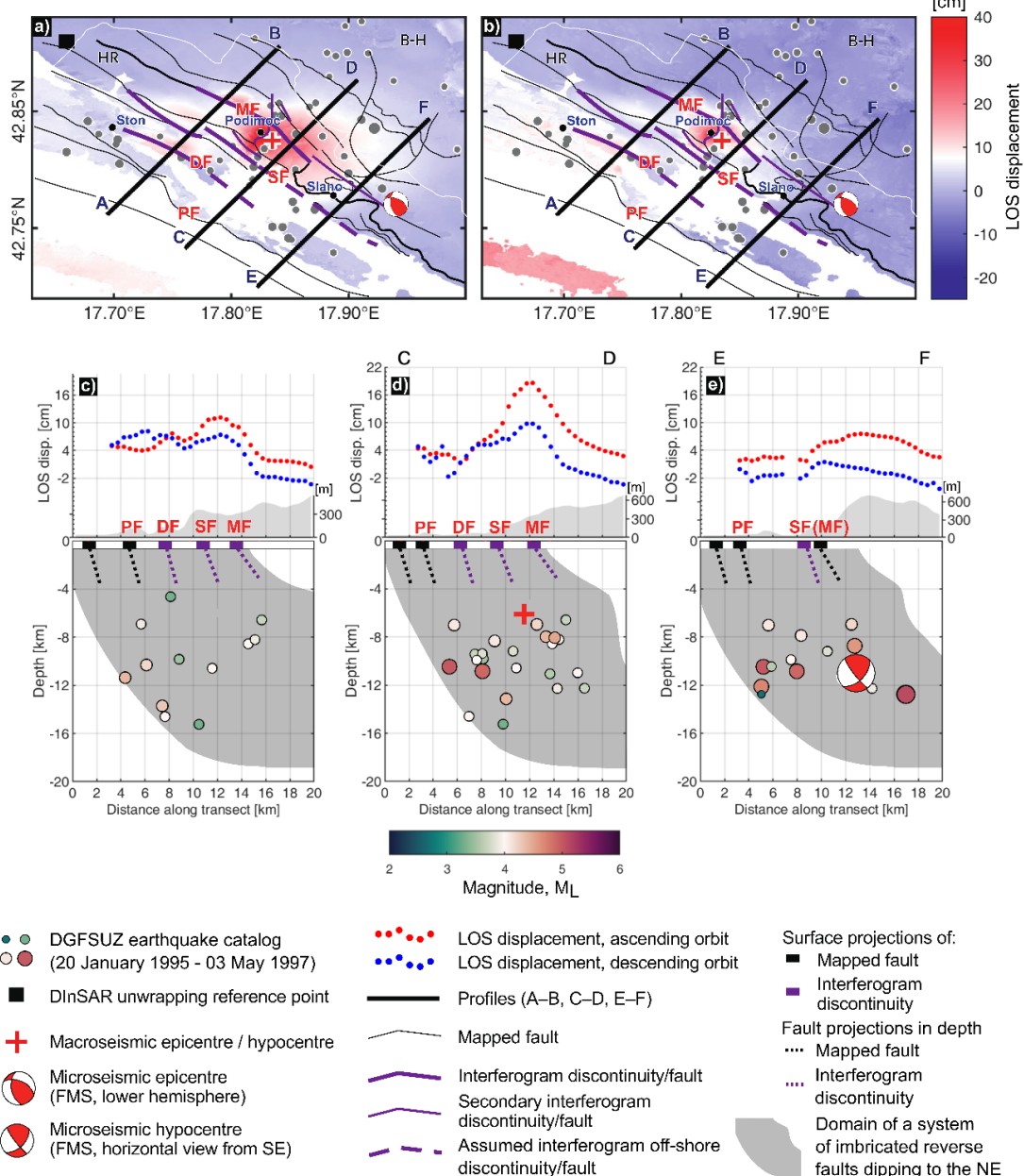

**Figure 13.** Three profiles A–B, C–D, E–F (black lines) over (**a**) ascending and (**b**) descending track coseismic interferograms, with relocated earthquakes and documented faults depicted with grey circles and black/purple lines, respectively. Only earthquake locations characterised by a standard error of the epicentre $\sigma_H \leq 5$ km and with azimuthal coverage characterised by a gap $\gamma \leq 150°$ are used. Subplots (**c**), (**d**), and (**e**) show DInSAR coseismic displacements, earthquake hypocentres, and points of cross-sections with the documented faults (PF, DF, SF, MF) 5 km around the profiles. Red and blue circles show the mean coseismic displacement from ERS-2 ascending and descending track interferograms along the profiles, respectively. Relocated earthquakes are shown by different colours according to their magnitude, and topography elevation is shown in light grey. Dark grey patch approximately shows the cross-section of the domain of imbricated reverse faults dipping to the NE. The FMS of the mainshock in (**a**) and (**b**) is shown as horizontal equal-area projection on the lower hemisphere. In (**e**) the view of the focal sphere is horizontally from SE, perpendicularly to the strike of the transect E–F.

After that, the rupture enters the area of the largest observed damage, characterised also by the largest cumulative ground uplift (Figure 7b,d). The very dense fringe pattern P5, partly decorrelated

(Figure 8a,b) suggests that most accumulated stress was released closer to the surface (in agreement with the macroseismic focal depth of 6 km), which (according to [53]) can result with potential surface rupture and secondary effects, e.g., liquefaction, building damage, etc. As the earthquake occurred 24 years ago, it is not possible to confirm the cause of the decorrelation in the macroseismic epicentral area. Interferograms in this area strongly indicate that several imbricated reverse faults (e.g., the Mravinca Fault; see Figures 9 and 13) were activated by the main rupture. The most clear case is the one of the fringe discontinuity D3a associated with pattern P3 (Figures 7 and 8a; maximum LOS displacements of 19 and 14 cm in ascending and descending interferograms, respectively) and identified in the field as a NW-striking reverse Mravinca fault, an approximately 10 km long segment of the regionally significant High Karst Nappe (MF in Figure 9). Similarly, the smaller or less clearly expressed ruptures (short dashes in Figure 8a) that cannot be attributed to any known aftershock (or indeed to any of the mapped faults), were also probably activated by the main event. Some of them strike at large angles or even perpendicularly to the main mapped faults, indicating complex stress field, and activation of a set of smaller linked faults, i.e., conjugate faults within the system. A similar observation related to the Ridgecrest earthquake was published by Ross et al. [54].

Most probably the main rupture continued further for several kilometres along the fault towards Ston (segment D2 in Figure 8a, and the associated pattern P2 with the maximum displacement of about 10 cm), where the final stress release occurred. The rupture of this segment of the Slano fault combined with the soil amplification (see above) could explain the severe damage in Ston, which is 21 km away from the microseismic epicentre.

Almost all of the observed DInSAR coseismic displacements are positive (upwards) and confined to the hanging walls of imbricated reverse faults (Figures 9 and 13), suggesting coseismic relaxation of deformation accumulated in the faults' hanging walls by tectonic forces related to the compressional/transpressional stress field, characterised by dominant NE–SW (locally NW–SE) oriented shortening. Virtually complete lack of aftershocks in the most shaken area (outlined in Figure 14), as well as very weak seismicity since 1996 (see Figure 4b) imply that the stress release by the mainshock was nearly total.

The expected subsurface rupture length ($L$) for an $M_w$ 6.0 earthquake is between 11.5 and 14.0 km (according to [52]), which roughly matches the length of the D1 line, as shown in Figure 8a. Assuming that the centroid location corresponds to the area of the largest observed displacement (Figure 7b,d), and considering the centroid time is 8.1 s later than the hypocentral time (DT for event No. 1 in Table 1), an average rupture velocity of about 1.5 km/s is implied, which is lower than the global average for such events (e.g., [55]).

As shown above, there is a strong possibility that several faults were activated during the main rupture, which considerably increases the total rupture length. The scalar seismic moment $M_o = A\mu u$ (where $A$—rupture area, $\mu = 3.2 \times 10^{10}$ Pa—shear modulus of the crust, $u$—average slip on the fault) of $1.2 \times 10^{18}$ Nm as given in the Global CMT Catalogue [30,31,33], implies the product $A \times u = L \times W \times u = 3.8 \times 10^7$ m$^3$ (where $L$, $W$—average rupture length, and width, respectively). Tentatively assuming $L = 25$ km along all activated faults and subfaults, and reasonably assuming the mean slip $u$ in the range of 15–25 cm led to estimates of $W$ between 6 and 10 km. As the earthquake started at the depth of about 11 km, and the fringe patterns P1a and P1b (Figure 8a) extend well away from the fault, it seems reasonable to assume that most of the slip in the initial part of the rupture was confined to the deeper parts of the fault. Saturated and concentrated fringes in patterns P3 and especially P5 suggest that in the later stages the rupture moved towards the surface, where it also activated a number of smaller faults (Figures 8a and 9).

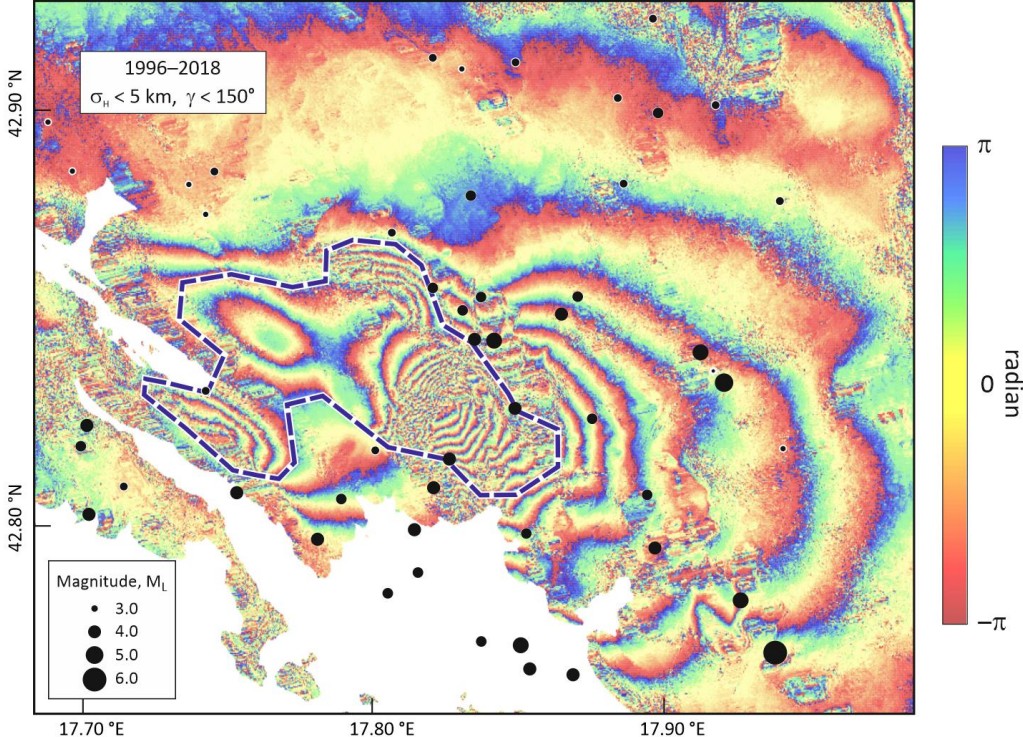

**Figure 14.** Epicentres of reliably located earthquakes of the Ston–Slano sequence overlain on the ascending orbit interferogram from Figure 7a. Only locations characterised by a standard error of the epicentre $\sigma_H \leq 5$ km and with azimuthal coverage characterised by a gap $\gamma \leq 150°$ are shown. Thick blue dashed line encloses the most shaken area which is virtually aftershock-free.

Locations of foci (Figures 4 and 13), as well as the DInSAR interferograms suggest that the accumulated stress during the Ston–Slano sequence was released by a complex rupture of the system of imbricated faults (at least the faults PF, DF, SF, and MF; see Figure 9) in the length of more than 20 km. As outlined above, the large part of the overall rupture probably occurred during the mainshock itself. Such a complicated coseismic faulting pattern has never before been documented in the Dinarides. However, the Ston–Slano earthquake is the only event in the Dinarides large enough to be analysed by the DInSAR technique, which suggests that multiple faulting and complex surface deformation may be common in the area, but may not be always detectable or measurable. If so, this can present a serious problem in defining realistic hazard scenarios, especially in deterministic hazard assessment.

## 6. Conclusions

This multidisciplinary study revealed a complex pattern of faulting and rupture propagation during the mainshock of the Ston–Slano earthquake sequence (5 September, 1996, $M_w = 6.0$). In spite of rather scarce seismological data of that time, we were able to identify the faults that ruptured in the first week of the sequence, most probably during the main rupture itself. The fault-plane solution and the microseismic hypocentre suggest that the reverse rupture with a strong left-lateral component probably started on the Slano fault at the depth of about 11 km. Proceeding unilaterally to the NW with the velocity of about 1.5 km/s, the rupture propagated for about 11 km and entered the area of the maximum stress release surrounding the villages of Mravinca and Podimoć, where the highest intensities were observed. DInSAR interferograms suggest that several hanging wall faults were activated here, the largest of which is the Mravinca reverse fault as a segment of the regional High Karst Nappe. Some of those smaller dislocations strike at large angles to the prevailing strike (SE–NW) of major structures. Continuing further towards Ston, the final stress was released as far as 20 km away from the epicentre, about three km to the NE of Ston. The DInSAR interferograms suggest that

coseismic ground deformation is observed in the area of about 200 km$^2$, with the maximum LOS ground displacement reaching 38 cm.

This area of the southern External Dinarides is characterised by NE–SW trending regional shortening that is partitioned along the closely spaced imbricated reverse fault system. In this imbricated fault system, large dislocation along the most stressed fault surface is likely to induce stress release also within the already stressed faults, mostly in the hanging wall of the primary rupture. If multiple faulting processes of strong events as complex as the one described above are a rule in the area, rather than exception, this could present a serious problem in defining realistic hazard scenarios, especially in deterministic hazard assessment.

**Author Contributions:** Conceptualization of research, M.G., M.H., B.M. and I.V.; data curation, M.G., M.H., B.M. and I.V.; DInSAR investigations, M.G.; seismological investigations, M.H.; geological investigations, B.M. and I.V.; software, M.G., M.H. and B.M.; visualization, M.G., M.H., B.M. and I.V.; writing—original draft, M.G., M.H., B.M. and I.V.; writing—review and editing, M.G., M.H., B.M. and I.V.; funding acquisition, B.P. and M.H.; project administration. B.P. and M.H. All authors have read and agreed to the published version of the manuscript.

**Funding:** This research was funded by the Croatian Science Foundation under grants IP-01-2018-8944 and IP-2014-09-9666.

**Acknowledgments:** We acknowledge the free access to satellite images provided by the European Space Agency, which enabled us to perform DInSAR investigation of the Ston–Slano earthquake sequence. We also thank the three anonymous reviewers for their very useful comments, which helped us improve the manuscript.

**Conflicts of Interest:** The authors declare no conflict of interest.

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
