# Peer review of "Constraints on Complex Faulting during the 1996 Ston–Slano (Croatia) Earthquake Inferred from the DInSAR, Seismological, and Geological Observations"

_remotesensing, doi:10.3390/rs12071157_

Round 1

Reviewer 1 Report

This paper is generally well-written. The authors utilized a variety of datasets ranging from seismological and geological to the geodetic observation, i.e. InSAR. They have conducted a comprehensive research about the faulting structure related to the Ston–Slano earthquake which is a useful reference for seismic hazard assessment. I am also impressed by the logical flow of the writing which is scientifically sound. Therefore, I would only ask for a minor correction before accepting for publication.

Main comments:

1 In the introduction, the authors didn’t seem to address their research importance enough. They need to highlight why these observations are important and why the faulting system needs to be re-analyzed, given the fact that it has been studied several times previously.

2 Although the authors used InSAR observations and emphases its importance of identifying ruptured fault segments, they did not model the fault slip as they did in FMS. It may be useful to try different fault slip models to fit the InSAR observation so that a best-fit solution can be found which may explain the fault system better.

Some other minor comments:

Table 2, the unit for Btemp should be days.

Figure 12, would the colour bar be [-2.88 to 2.88]?

Figure 14: missing colour bar. Missing also the type of the earthquake magnitude (MS OR MW)

Best to use consistent sub-headings for all figures. For example, you use ‘a), b)’ for Figure 1, but ‘a, b’ for Figure 7, then none in Figure 10

Need to give the date, location, and photographer of the photos in Figure 1 and 11.

Reviewer 2 Report

Very good contribution, which successfully combines various methods to get a comprehensive analysis of the event.

In line 235, citation [41] should be given instead of misquote [42].

Reference [34] is not mentioned anywhere in the article, and should therefore be eliminated.

Wrong text formatting on lines 323 to 349, 490 to 511.

Reviewer 3 Report

This is a really interesting case study documenting the use of Differential InSAR combined with, geological observations and seismic records/focal mechanisms to reveal the complex pattern of ground deformation caused by the 1996 Ston–Slano earthquake in Croatia. The application of DInSAR to understanding of the deformation field caused by an earthquake is not new, nor is the approach used here, but it seems novel in this region. There is clearly other which could be done to provide further answers, this is still a really interesting and compelling story and has important implications for seismic geohazard assessment in this region.

The paper is in general very well written with only a few places where some additional punctuation is required or acronyms not defined for example. There are however quite a few places where the explanations or descriptions of concepts or techniques are rather clumsily or confusingly or inconsistently worded. These I have highlighted in yellow and annotated in the attached pdf - I felt this was slightly easier than making a separate list of page numbers and suggestions - I hope this is helpful.  So I have made suggestions where the wording/explanation of something is not clear or is confusing to the reader. Plus suggested improvements to figures where symbology is ambiguous or obscures the details. 

I feel the importance of the demonstration of deformation complexity in this case could be emphasised further
